# Low birth weight, household socio-economic status, water and sanitation are associated with stunting and wasting among children aged 6–23 months: Results from a national survey in Ghana

Hammond Yaw Addae[1,2]*, Mohammed Sulemana[1], Taminu Yakubu[2,3], Ambrose Atosona[2,4], Rafatu Tahiru[2,5], Fusta Azupogo[6]

1 Nursing & Midwifery Training College, Kpembe, Salaga, Ghana, 2 Department of Biochemistry and Biotechnology, College of Science, Kwame Nkrumah University of Science and Technology, Kumasi, Ghana, 3 Department of Nutrition & Dietetics, Tamale Technical University, Tamale, Ghana, 4 Department of Nutritional Sciences, School of Allied Health Sciences, University for Development Studies, Tamale, Ghana, 5 Community Health Nurse Training College, Tamale, Ghana, 6 Faculty of Agriculture, Food and Consumer Sciences, Department of Family and Consumer Sciences, University for Development Studies, Tamale, Ghana

* hamondd@yahoo.com

**Data Availability Statement:** The data for this study are publicly available at https://mics.unicef.

## Abstract

### Background

Stunting and wasting are key public health problems in Ghana that are significantly linked with mortality and morbidity risk among children. However, information on their associated factors using nationally representative data is scanty in Ghana. This study investigated the influence of Infant and Young Child Feeding (IYCF) indicators, socio-demographic and economic related factors, and water and sanitation on stunting and wasting, using nationally representative data in Ghana.

### Methods

This is a secondary data analysis of the most recent (2017/2018) Ghana Multi-Indicator Cluster Survey (MICS) datasets. The multi-indicator cluster survey is a national cross-sectional household survey with rich data on women of reproductive age and children under the age of five. The survey used a two-stage sampling method in the selection of respondents and a computer-assisted personal interviewing technique to administer structured questionnaires from October 2017 to January 2018. The present study involved 2529 mother-child pairs, with their children aged 6 to 23 months. We used the Complex Sample procedures in SPSS, adjusting for clustering and stratification effects. In a bivariate logistic regression, variables with P-values ≤ 0.05 were included in a backward multivariate logistic regression to identify the significant factors associated with stunting and wasting.

org/surveys. Interested researchers can reproduce
our study's results and findings in its entirety by
obtaining directly these datasets from UNICEF's
MICs Program and replicating the protocol in our
methods. The access granted authors are
applicable to all other interested researchers, on
request from UNICEF's online repository.

**Funding:** The authors received no specific funding
for this work.

**Competing interests:** The authors have declared
that no competing interests exist.

## Results

The mean age of children was 14.32 ± 0.14 months, with slightly more being males (50.4%). About 12% and 16% of the children were wasted and stunted, respectively. There were 39.4%, 25.9%, and 13.7% of children who, respectively, satisfied the minimum meal frequency (MMF), minimum dietary diversity (MDD), and minimum acceptable diet (MAD). None of the IYCF indicators was significantly associated with stunting or wasting in the multivariate analysis but low socio-economic status, low birth weight, being a male child and unimproved toilet facilities were significantly associated with both wasting and stunting.

## Conclusion

Our findings suggest that aside from the pre-natal period, in certain contexts, household factors such as low socio-economic status and poor water and sanitation, may be stronger predictors of undernutrition. A combination of nutrition-specific and nutrition-sensitive interventions including the pre-natal period to simultaneously address the multiple determinants of undernutrition need strengthening.

## Introduction

Globally, the most significant singular driver of mortality and morbidity is sub-optimal diet [1]. It is a well-known fact that undernutrition has retarding effects on all aspects of human development and is particularly profound among children of south-east Asia and Sub-Saharan Africa [2,3]. However, its causes are region-specific and complex. The World Health Organization estimates that 144 million, 47 million and 38 million children under five (5) years are stunted, wasted and overweight, respectively [4]. It is also estimated that 40% and 30% of the world's stunted and wasted children, respectively, live in Africa alone [4], with West Africa, in particular, having substantially higher rates [3]. In Ghana, however, although undernutrition rates are on a downward trajectory, regional disparities still exist [5,6], and current rates are still high relative to internationally acceptable thresholds [7,8]. Nationally representative estimates classify 18% of children under 5 years as stunted, 7% as wasted and 66% as anaemic, with 1 in every 19 children likely to die before age five (5) [6,9].

The ramifications and harm caused by undernutrition, especially during the first 1000 days in a child's life and its long-term effects on economic productivity, educational achievements and overall morbidity and mortality have been documented extensively [10–12]. In Ghana nonetheless, a myriad of research works have implicated sub-optimal Infant and Young Child Feeding (IYCF) practices [13,14]; poor wealth distribution [15]; low maternal education [13]; low birth weight [14]; and inadequate sanitary conditions [13] as plausible causes of malnutrition.

Despite a well-established epidemiology of malnutrition in Ghana, the lack of prioritization has led to interventions not yielding the desired impact [16]. The infant feeding landscape in Ghana has been described as one saddled with decreasing exclusive breastfeeding rates [17], increasing reliance on commercial infant formula/cereals [18], and extensive use of monotonous staples with low micronutrient content [19]. These have led to low prevalence of IYCF indicators such as Minimum Meal Frequency (MMF), Minimum Dietary Diversity (MDD), and Minimum Acceptable Diet (MAD). For instance, a Ghanaian national survey analysis

found that 42% of children aged 6 to 23 months meet the recommended MMF, with 38% meeting the MDD and only 14% the MAD [20].

Promotion of nutrition-specific interventions which target immediate causes of malnutrition, such as sub-optimal dietary practices, under these contexts could mean improving malnutrition [21,22]. To a larger extent, it could also aid in ensuring optimal childhood growth, enhance cognitive development [10], reduce malnutrition rates and potentially protect children against opportunistic infections by strengthening their immune systems [23,24]. This assertion that improving dietary factors would inevitably improve malnutrition rates may only sometimes be true, as the link between sub-optimal diet and malnutrition could be amplified by other factors [25,26]. Besides, malnutrition itself could be caused by other factors that are not diet-related. Consequently, it is being advocated that the addition of nutrition-sensitive interventions that target the underlying causes of malnutrition may be necessary and synergistic to nutrition-specific interventions if improved nutrition outcomes are to be achieved and sustained [27–29].

The multidimensional nature of the IYCF-undernutrition nexus suggests it could be complicated by extraneous factors peculiar to sub-Saharan Africa such as poor water sources [30], inadequate sanitary conditions [25,31], malarial infections [32], low socio-economic status [33] and socio-cultural norms and practices [34].

There is sparse literature exploring factors associated with both wasting and stunting using nationally representative data in Ghana. As a result, the purpose of this study was to investigate the relationship between IYCF indicators, socio-economic, demographic, water and sanitation factors, and the outcome variables stunting and wasting among children aged 6 to 23 months. Limiting the outcome variables to only wasting and stunting is in tandem with the SDG approach [35] and mainly because underweight is a composite measure of both stunting and wasting. This study would synthesize these factors of undernutrition and recommend approaches that may reduce malnutrition in the wake of Sustainable Development Goal 2.2, which seeks to reduce stunting and wasting in children under five (5) years to internationally acceptable levels by 2025 and end all forms of malnutrition by 2030 [35].

## Materials and methods

### Study design, site, and population

This is a secondary data analysis of the most recent (2017/2018) Ghana Multi-Indicator Cluster Survey (MICS) datasets. The Ghana MICS, from which the data for this study was based was a national cross-sectional household survey undertaken in Ghana by the Ghana Statistical Service in collaboration with stakeholders and development partners, to gather data on a wide range of indicators on the situation of children, and women and men aged 15–49 years. The present secondary analysis focused on mother-child pairs, with their children aged 6 to 23 months. The MICS survey provides statistically sound data for policy formulation, monitoring progress and evaluation of existing programs' impact [36].

### Sampling and sample size

The Ghana MICS 2017/18 used a two-stage sampling method which comprised a selection of 660 enumeration areas using probability proportionate to household size. The second-stage sampling employed a systematic random sampling to select 20 households in each Electoral Area (EA), resulting in 13,200 households across all ten (10) regions of Ghana. The total number of individuals in all the selected households for the Ghana MICS 2017/18 survey was 60,581, and those eligible for interview were 37,951. Out of these, 8,903 were aged 0 to 5 years. Specific details of the sampling methods have been described elsewhere [36]. Upon excluding

non-response, children aged 24 months to 5 years, children less than six (6) months and those with missing anthropometry (age, weight, and height), 2,529 infants aged 6 to 23 months, each paired with their mother, remained for inclusion in the present study. The study included only 6 to 23 months children partly because the IYCF indicators, which are key aspects of this study, are only applicable among children within this age group. The flow chart showing the detailed selection of the population for analysis is shown in Fig 1.

## Study variables

**Dependent variables.** The dependent variables were stunting and wasting. Children with z-scores less than -2 SD relative to the 2007 WHO reference population median value [37] were considered stunted and wasted for height-for-age and weight-for-age, respectively. And children with z-scores equal to or more than -2 SD were considered not stunted and not wasted, respectively.

**Independent variables.** The independent variables were MDD, MMF, MAD, age of child, sex of child, mother's age, mother's education, place of residence, child history of fever in the past 2 weeks, timely prenatal care, child's birth weight, sex of household head, wealth index, household source of drinking water and type of toilet facility.

## Data collection

The survey (MICS 2017/18) used a computer-assisted personal interviewing technique to administer pre-tested structured questionnaires to eligible respondents spanning October 2017 to January 2018. The survey employed six questionnaires: household questionnaire, water quality testing questionnaire, men's questionnaire, women's questionnaire, under-5 children's questionnaire and 5–17 years children's questionnaire. In the present study, data gathered from the water and sanitation and household characteristics modules of the household questionnaire, women's background module of the women's questionnaire and child's background, anthropometry and breastfeeding and dietary intake modules of the under-5 children's questionnaire were used. Details of the data collection methods have been published elsewhere [36]. The WHO IYCF indicators were validated for children aged 6–23 months, which is why we limited the current investigation to this age group.

## Assessment of stunting and wasting

Stunting and wasting are considered the two most common indices for defining undernutrition [38]. This study used a combination of children's height, weight, sex, and age to produce anthropometric indices. Height and age were used to calculate height-for-age z-scores (HAZ), which connotes stunting status, a measure of chronic malnutrition. Weight and height were also used to calculate weight-for-height z-scores (WHZ), which reflect wasting status, a measure of acute malnutrition. Chronic malnutrition is often due to prolonged inadequate dietary intake and illness, while acute malnutrition is usually due to food deprivation and illness of recent origin [39].

## Assessment of infant and young child feeding indicators

These indicators were measured using the 24-hour dietary recall method, and included MDD, MMF and MAD [40]. These indicators have been validated within the developing country context as reliable indicators for predicting access to food, overall quality of diet and nutritional status of children aged 6 to 23 months [41].

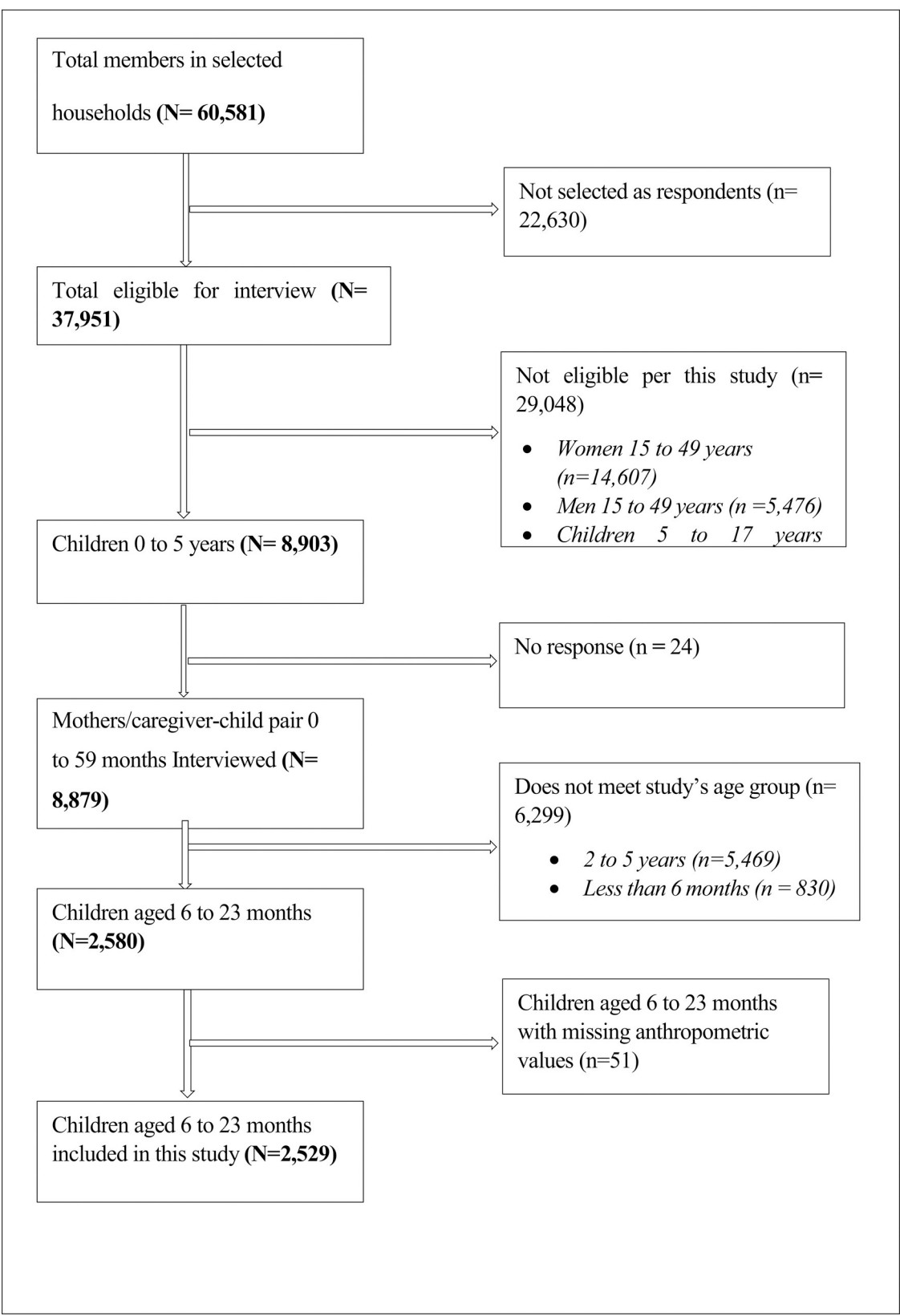

**Fig 1. Flow chart showing selection of children 6 to 23 months from MICS 2017/18.**

To measure these indicators, this study categorised the specific foods consumed by children over the past 24 hours to conform with the WHO 8-food group indicator [40]. These comprised (1) grains, tubers, and roots, (2) dairy products, (3) meat and fish, (4) eggs (5) legumes, nuts, and seeds, (6) dark green leafy vegetables and vitamin A-rich foods, (7) other fruits and vegetables and (8) breastmilk. The operational definitions of IYCF indicators were adopted from WHO/UNICEF [40].

MDD is a WHO-recommended indicator for measuring diet quality. Children aged 6 to 23 months are deemed to have met the MDD if they ate foods from five (5) or more food groups in the past 24 hours. MMF is an age-specific indicator that measures food quantity and frequency. Regarding breastfed children, those aged 6 to 8 months who have consumed two or more solid, semi-solid, or soft feeds are deemed to have met the MMF, while those aged 9 to 23 months need three (3) or more solid, semi-solid, or soft feeds to meet the MMF. For non-breastfed children, those aged 6 to 23 months need four (4) or more solid, semi-solid, or soft feeds to meet the MMF. Regarding MAD, children were deemed to have met this indicator if they have met both MDD and MMF.

## Assessment of water and sanitation systems and wealth index

The water and sanitation systems were assessed based on the recommendations of WHO and UNICEF [42]. In the present study, piped water, public tap, borehole, protected well, protected springs, rainwater in a tank, and bottled or sachet water were classified as protected water, while, river, irrigation channel, lake, dam, pond, stream, unprotected well, or unprotected spring water were categorized as unprotected water. For sanitation, flush toilets to sewer system, flush toilets to septic tank, ventilated improved pit latrine, pour-flush latrines to sewer system, pour-flush latrines to septic tanks, composting toilets, pit latrine with slab were classified as improved facilities, while flush toilets to open drain, pit latrine without slap, bucket latrine, hanging toilet or hanging latrine was classified as unimproved facilities. Open defecation was defined as households without toilet facilities or households that defecate in the bush or fields. Wealth index was computed using the principal component analysis, where household assets were used as bases for determining wealth. Wealth index was classified as low, medium or high. Specific details of computations of wealth index of the Ghana MICS 2017/2018 are described in the published report [36].

## Quality control measures

The Ghana MICS 2017/18 employed several on-field methods to reduce bias and minimize error. According to the published report [36], field workers were trained extensively on interviewing techniques and questionnaire administration and practised role-play interviews. This included five (5) days in on-field practice and one (1) day in a pilot survey to be conversant with the questionnaire administration process. Field workers responsible for taking anthropometric measurements and samples for testing were trained for an additional 14 days and six (6) more days for pilot and on-field practice. Monitoring and supervision of fieldwork were carried out by team supervisors. To ensure high standards, team supervisors observed the interviewers' skills daily and re-interviewed one household in each cluster using purposive and random sampling procedures. These are contained in the published report of MICS 2017/18 [36].

## Statistical analysis

This study's statistical analysis was conducted using SPSS version 22 (SSPS Inc. Chicago, IL, USA). Descriptive statistics were used to present data on the study population's socio-demographic characteristics. Bivariate analysis was performed using chi-square ($\chi$2) test and binary logistic regression analysis to determine the association between the dependent and

independent factors. The independent factors that were significant with p < 0.05 in the bivariate analysis were considered for multivariable logistic regression analysis [43]. Furthermore, given that the MICS used a multistage cluster study design which deviates from the principle of equal probability in simple random sampling, clustering and stratification were adjusted for in all analyses using SPSS complex samples module procedure. This involved creating a complex samples plan file (csplan) using strata, cluster, and sample weight variables in our combined dataset. The csplan file served as the bases for all analyses in this study. The intent was to normalise the variance and eliminate the risks of bias towards under and over-sampled sub-populations. All analyses were carried out with 95% Confidence Interval (CI), and factors were deemed statistically significant if their p-values were less than 0.05 in both the bivariate and multivariable analyses. The model fitness test was determined based on chi-square and overall model p-value using Hosmer Lemeshow test.

### Ethical consideration

The Ghana Statistical Service sought and were granted ethical approval for the survey (2017/18 MICS) from the Ghana Health Service Ethical Review Committee. Participants' verbal consent was sought. Consent was obtained from parents/caretakers of participants younger than 18 years. They were made aware of the voluntary nature of the participation, as well as the confidentiality and anonymity of the data. In so doing, datasets were anonymized and authors had no access to individual data identifiers. Participants were also made aware of their freedom to discontinue the interview at any moment, as well as their right not to respond to any or all questions.

## Results

### Characteristics of the study population

Table 1 depicts the socio-economic and demographic characteristics of the participants. The mean ages of children and mothers were found to be 14.32 ± 0.14 months (95% CI: 14.05–14.59) and 29.37 ± 0.31 years (95% CI: 28.76–29.98), respectively. The sex of children were of equal proportion. However, a larger proportion of mothers were middle-aged (42.8%), had low educational status (60.2%), were currently married (79.7%), and were health insured (53.7%). Regarding households of these mother-child pairs, 56.6% were classified as rural, and 41.1% had a low wealth index. Most (24.0%) of the households were in the Ashanti Region, with the Upper West Region having the least (2.3%).

### Food groups consumption, IYCF indicators and nutritional status of children

The prevalence of wasting and stunting among the children was 12.1% and 15.9% respectively. Consumption data for the eight (8) food groups indicates that the majority (80.1%) of children ate from group 1 (grains, tubers, and roots), with less than 46% of children eating from the remaining 6 food groups in the past 24 hours (Table 2). In all, the average number of food groups children had eaten from was 3.37 ± 0.05. For the IYCF indicators, 39.3% of the children met the age-appropriate MMF and a much lower percentage of 25.9% and 13.7% met the MDD and MAD, respectively.

Stratifying nutritional status by IYCF indicators and food groups consumed (Table 2) revealed that only MDD was statistically significant with wasting but not stunting. Children who met MDD were significantly less likely to be wasted as compared to those who did not meet MDD (9.0% vs 13.2%, p = 0.044). MMF, MDD and MAD were not significantly associated with stunting. Children that were still breastfeeding were significantly less likely to be stunted as

**Table 1. Socio-economic and demographic characteristics of study population (n = 2529).**

| Characteristic | Frequency (n = 2529) | Percentage (%) |
|---|---|---|
| **Age of child (months)\*** | | |
| 6 to 11 | 862 | 34.1 |
| 12 to 17 | 811 | 32.1 |
| 18 to 23 | 856 | 33.8 |
| **Sex of child** | | |
| Male | 1274 | 50.4 |
| Female | 1255 | 49.6 |
| **Mother's age (years)\*\*** | | |
| < 25 | 759 | 30.0 |
| 25 to 34 | 1082 | 42.8 |
| >34 | 688 | 27.2 |
| **Mother's education** | | |
| None (No education) | 562 | 22.2 |
| Low (Primary & JHS) | 1523 | 60.2 |
| High (SHS & above) | 444 | 17.6 |
| **Marital status** | | |
| Married | 2016 | 79.7 |
| Not Married | 513 | 20.3 |
| **Area of residence** | | |
| Rural | 1431 | 56.6 |
| Urban | 1098 | 43.4 |
| **Wealth** | | |
| Low | 1038 | 41.1 |
| Medium | 472 | 18.6 |
| High | 1019 | 40.3 |
| **Administrative region** | | |
| Western | 290 | 11.5 |
| Central | 246 | 9.7 |
| Greater Accra | 237 | 9.4 |
| Volta | 209 | 8.3 |
| Eastern | 286 | 11.3 |
| Ashanti | 608 | 24.0 |
| Brong Ahafo | 231 | 9.1 |
| Northern | 281 | 11.1 |
| Upper East | 82 | 3.2 |
| Upper West | 59 | 2.3 |
| **Health insurance status** | | |
| Yes | 1359 | 53.7 |
| No | 1170 | 46.3 |
| **Place of delivery** | | |
| Home | 465 | 18.4 |
| Private hospital | 344 | 13.6 |
| Government hospital | 1720 | 68.0 |

\*Mean child age ± standard error = 14.32 ± 0.14 months (95% Confidence Interval: 14.05–14.59).

\*\*Mean maternal age ± standard error = 29.37 ± 0.31 years (95% Confidence Interval: 28.76–29.98).

JHS = Junior High School; SHS = Senior High School.

**Table 2. Wasting and stunting stratified by IYCF indicators and food groups consumed.**

| Characteristic | Frequency (%) | Not wasted n (%) | Wasted n (%) | p-value | Not Stunted n(%) | Stunted n(%) | Test statistics, p-value |
|---|---|---|---|---|---|---|---|
| **Total** | 2529(%) | 2223(87.9) | 306(12.1) | | 2126(84.1) | 403(15.9) | |
| **Minimum meal frequency** | | | | | | | |
| Yes (>4 food groups) | 995(39.3) | 881(88.5) | 114(11.5) | p = 0.62 | 847(85.1) | 148(14.9) | χ2 = 1.3, p = 0.390 |
| No (< 5 food groups) | 1534(60.7) | 1342(87.5) | 192(12.5) | | 1279(83.4) | 255(16.6) | |
| **Minimum dietary diversity** | | | | | | | |
| Yes | 656(25.9) | 597(91.0) | 59(9.0) | **p = 0.044** | 546(83.2) | 110(16.8) | χ2 = 0.6, p = 0.560 |
| No | 1873(74.1) | 1626(86.8) | 247(13.2) | | 1580(84.2) | 293(15.6) | |
| **Minimum acceptable diet** | | | | | | | |
| Yes | 346(13.7) | 312(90.2) | 34(9.8) | p = 0.32 | 287(82.9) | 59(17.1) | χ2 = 0.5, p = 0.620 |
| No | 2183(86.3) | 1911(87.5) | 272(12.5) | | 1839(84.2) | 344(15.8) | |
| **Currently breastfeeding** | | | | | | | |
| Yes | 1969(77.9) | 1710(86.8) | 259(13.2) | **p = 0.03** | 1686(85.6) | 283(14.4) | **χ2 = 15.4, p = 0.007** |
| No | 560(22.1) | 513(91.6) | 47(8.4) | | 440(78.6) | 120(21.4) | |
| **Grains, tubers and roots** | | | | | | | |
| Yes | 2025(80.1) | 1784(88.1) | 241(11.9) | p = 0.73 | 1683(83.1) | 342(16.9) | **χ2 = 7.9, p = 0.036** |
| No | 504(19.9) | 439(87.1) | 65(12.9) | | 443(87.9) | 61(12.1) | |
| **Dairy products** | | | | | | | |
| Yes | 534(21.1) | 489(91.6) | 45(8.4) | **p = 0.038** | 452(84.6) | 82(15.4) | χ2 = 0.1, p = 0.790 |
| No | 1995(78.9) | 1734(86.9) | 261(13.1) | | 1674(83.9) | 321(16.1) | |
| **Meat and fish** | | | | | | | |
| Yes | 1142(45.2) | 1032(90.4) | 110(9.6) | p = 0.06 | 942(82.5) | 200(17.5) | χ2 = 4.0, p = 0.180 |
| No | 1387(54.8) | 1191(85.9) | 196(14.1) | | 1184(85.4) | 203(14.6) | |
| **Eggs** | | | | | | | |
| Yes | 520(20.6) | 486(93.5) | 34(6.5) | **p = 0.002** | 450(86.5) | 70(13.5) | χ2 = 3.2, p = 0.210 |
| No | 2009(79.4) | 1737(86.5) | 272(13.5) | | 1676(83.4) | 333(16.6) | |
| **Legumes, nuts and seeds** | | | | | | | |
| Yes | 422(16.7) | 381(90.3) | 41(9.7) | p = 0.22 | 336(79.6) | 86(20.4) | **χ2 = 7.0, p = 0.034** |
| No | 2107(83.3) | 1842(87.4) | 265(12.6) | | 1790(85.0) | 317(15.0) | |
| **Dark green leafy vegetables and vitamin A-rich foods** | | | | | | | |
| Yes | 1098(43.4) | 980(89.3) | 118(10.7) | p = 0.22 | 889(81.0) | 209(19.0) | **χ2 = 13.4, p = 0.008** |
| No | 1431(56.6) | 1243(86.9) | 188(13.1) | | 1237(86.4) | 194(13.6) | |
| **Other fruits and vegetables** | | | | | | | |
| Yes | 854(33.8) | 779(91.2) | 75(8.8) | **p = 0.018** | 700(82.0) | 154(18.0) | χ2 = 4.3, p = 0.100 |
| No | 1675(66.2) | 1444(86.2) | 231(13.8) | | 1426(85.1) | 249(14.9) | |

All food groups (mean ± standard error) = 3.37 ± 0.05 [C.I 95% = 3.27–3.47].

compared to children that were not (14.1% vs 21.4%, p = 0.007). Those that ate dairy products (8.4% vs 13.1%, p = 0.038), eggs (6.5% vs 13.5, p = 0.002), other fruits and vegetables (8.8% vs 13.8%, p = 0.018) were less likely to be wasted as compared to children that did not.

## Bivariate factors associated with wasting and stunting

In a bivariate logistic regression analysis as shown in Table 3, children that met MDD were 35% less likely to be wasted as compared to those that did not meet MDD (OR = 0.65, 95% CI: 0.43–0.99, p = 0.046). Male children were about 1.5 times more likely to be wasted (OR = 1.49, 95% CI: 0.87–2.57, p = 0.036) and 1.4 times more likely to be stunted (OR = 1.43, 95% CI:

**Table 3. Bivariate analysis of factors associated with wasting and stunting.**

| Variable | Wasting | | Stunting | |
|---|---|---|---|---|
| | Unadjusted OR (CI = 95%) | p-value | Unadjusted OR (CI = 95%) | p-value |
| **Minimum dietary diversity** | | | | |
| Yes (≥ 5 food groups) | 0.65(0.43–0.99) | **0.046** | 1.1(0.80–1.50) | 0.560 |
| No (< 5 food groups) | Ref. | | Ref. | |
| **Minimum meal frequency** | | | | |
| Yes | 0.90(0.59–1.38) | 0.633 | 0.89(0.66–1.19) | 0.413 |
| No | Ref. | | | |
| **Minimum adequate diet** | | | | |
| Yes | 0.77(0.46–1.30) | 0.330 | 0.90(0.59–1.36) | 0.603 |
| No | Ref. | | Ref. | |
| **Age of child (months)** | | | | |
| 6 to 11 | 1.49(0.87–2.57) | 0.150 | 2.94(2.06–4.19) | **<0.001** |
| 12 to 17 | 0.91(0.50–1.66) | 0.770 | 1.77(1.22–2.57) | **0.003** |
| 18 to 23 | Ref. | | Ref. | |
| **Sex of child** | | | | |
| Male | 1.47(1.03–2.10) | **0.036** | 1.43(1.06–1.94) | **0.019** |
| Female | Ref. | | Ref. | |
| **Mother's education** | | | | |
| None (No education) | 2.07(1.27–3.39) | **0.004** | 2.09(1.37–3.20) | **0.001** |
| Low (Primary & JHS) | 1.46(0.86–2.48) | 0.160 | 1.19(0.79–1.78) | 0.410 |
| High (SHS and above) | Ref. | | Ref. | |
| **Area of residence** | | | | |
| Rural | 1.05(0.71–1.53) | 0.820 | 1.39(1.01–1.90) | **0.043** |
| Urban | Ref. | | Ref. | |
| **Wealth** | | | | |
| Low | 1.56(1.02–2.38) | **0.039** | 1.49(1.05–2.12) | **0.025** |
| Medium | 0.80(0.45–1.40) | 0.431 | 0.99(0.63–1.55) | 0.958 |
| High | Ref. | | Ref. | |
| **Child ill with fever in last 2 weeks** | | | | |
| Yes | 1.81(1.21–2.71) | **0.004** | 1.48(1.05–2.07) | **0.023** |
| No | Ref. | | Ref. | |
| **Timely prenatal** | | | | |
| Yes (< 4 months) | Ref. | | Ref. | |
| No (4 months +) | 1.91(1.04–3.50) | **0.037** | 1.07(0.69–1.66) | 0.774 |
| **Child's birth weight** | | | | |
| Small | 3.25(1.33–7.98) | **0.010** | 2.14(1.11–4.12) | **0.023** |
| Average | 1.78(0.82–3.87) | 0.144 | 1.20(0.75–1.91) | 0.448 |
| Large | Ref. | | Ref. | |
| **Source of drinking water** | | | | |
| Unprotected | 1.97(1.07–3.62) | **0.029** | 1.80(0.94–3.42) | 0.074 |
| Protected | Ref. | | Ref. | |
| **Type of toilet facility** | | | | |
| Open defecation | 1.34(0.74–2.41) | 0.329 | 2.07(1.19–3.59) | **0.010** |
| Unimproved | 2.01(1.05–3.85) | **0.035** | 0.91(0.52–1.62) | 0.759 |
| Improved | Ref. | | Ref. | |
| **Sex of household head** | | | | |
| Male | Ref. | | Ref. | |

(*Continued*)

**Table 3.** (Continued)

| Variable | Wasting | | Stunting | |
|---|---|---|---|---|
| | Unadjusted OR (CI = 95%) | p-value | Unadjusted OR (CI = 95%) | p-value |
| Female | 1.39(0.59–3.28) | 0.451 | 0.58(0.35–0.94) | **0.026** |

Ref. = Reference category.

1.06–1.94, p = 0.019) as compared to their female counterparts. As compared to children of mothers with high educational status, children of mothers with low educational status were twice likely to be wasted (OR = 2.07, 95% CI: 1.27–3.39, p = 0.004) and twice likely to be stunted (OR = 2.09, 95% CI: 1.37–3.20, p = 0.001). Children that had a fever in the last two (2) weeks were about 1.5 times more likely to be wasted (OR = 1.81, 95% CI: 1.21–2.71, p = 0.004) and 1.1 times more likely to be stunted (OR = 1.48, 95% CI: 1.05–2.07, p = 0.023), as compared to children that did not have any fever. Also, children with small birth weight were 3.3 times and 2 times more likely to be wasted (OR = 3.25, 95% CI: 1.33–7.98, p = 0.010) and stunted (OR = 2.14, 95% CI: 1.11–4.12, p = 0.023) respectively, as compared to children with large birth weight. Lack of proper sanitation was a risk factor for both wasting and stunting as children of households with unimproved toilet facilities were 2 times more likely to be wasted (OR = 2.01, 95% CI: 1.05–3.85, p = 0.035) and households with open defecation were also 2 times more likely to be stunted (OR = 2.07, 95% CI: 1.19–3.59, p = 0.010) as compared to households with improved toilet facilities. Source of drinking water was only associated with wasting but not stunting; children in households with unprotected drinking water were 2 times more likely to be wasted (OR = 1.97, 95% CI: 1.07–3.62, p = 0.029) as compared to those with protected drinking water.

## Factors associated with wasting from multivariate analysis

The multivariable logistic regression analysis (Table 4) showed that MDD was not significantly associated with wasting. Mothers with low educational status were 2.8 times more likely to have children that were wasted than children of mothers with high educational status (AOR = 2.8, 95% CI: 1.16–6.77, p = 0.023). Further, as compared to female children, male children were 2.2 times more likely to be wasted (AOR = 2.19, 95% CI: 1.21–3.96, p = 0.009), and children that fell sick in the past 2 weeks were also 2.2 times more likely to be wasted (AOR = 2.15, 95% CI: 1.18–3.91, p = 0.012) relative to those that did not. Children with low and medium birth weight were about 5 times and 3 times more likely to be wasted respectively as compared to children with high birth weight (AOR = 4.84, 95% CI: 2.03–11.53, p < 0.001) and (AOR = 3.11, 95% CI: 1.74–5.58, p < 0.001). Lastly, children of households with unprotected sources of drinking water (AOR = 2.55, 95% CI: 1.22–5.34, p = 0.013) and unimproved toilet facilities (AOR = 2.16, 95% CI: 1.03–4.55, p = 0.042) were both about 2.5 times more likely to be wasted as compared to those that had protected source of drinking water and improved toilet facilities, respectively. Overall, these exposure factors accounted for 20.0% (Nagelkerke $R^2$ = 0.200) of the variance in wasting among 6 to 23 months children in Ghana.

## Factors associated with stunting from multivariate analysis

As shown in Table 5, male children were about 1.8 times more likely to be stunted as compared to their female counterparts (AOR = 1.80, 95% CI: 1.28–2.52, p = 0.001) and children from low wealth households were also about 2.2 times more likely to be stunted as compared to children from high wealth households (AOR = 2.18, 95% CI: 1.28–3.76, p = 0.005). Children with small

**Table 4. Factors associated with wasting among children aged 6 to 23 months.**

| Variable | Wasting | | |
|---|---|---|---|
| | Adjusted OR (AOR) | 95% Confidence Interval | | p-value |
| | | Lower | Upper | |
| **Sex of child** | | | | |
| Male | 2.19 | 1.21 | 3.96 | **0.009** |
| Female | Ref. | | | |
| **Mother's education** | | | | |
| None (No education) | 1.94 | 0.66 | 5.70 | 0.230 |
| Low (Primary & JHS) | 2.80 | 1.16 | 6.77 | **0.023** |
| High (SHS and above) | Ref. | | | |
| **Wealth** | | | | |
| Low | Ref. | | | |
| Medium | 0.32 | 0.13 | 0.81 | **0.016** |
| High | 0.56 | 0.23 | 1.36 | 0.201 |
| **Child ill with fever in last two (2) weeks** | | | | |
| Yes | 2.15 | 1.18 | 3.91 | **0.012** |
| No | Ref. | | | |
| **Child's birth weight** | | | | |
| Small | 4.84 | 2.03 | 11.53 | **<0.001** |
| Average | 3.11 | 1.74 | 5.58 | **<0.001** |
| Large | Ref. | | | |
| **Source of drinking water** | | | | |
| Unprotected | 2.55 | 1.22 | 5.34 | **0.013** |
| Protected | Ref. | | | |
| **Type of toilet facility** | | | | |
| Open defecation | 1.61 | 0.60 | 4.37 | 0.346 |
| Unimproved | 2.16 | 1.03 | 4.55 | **0.042** |
| Improved | Ref. | | | |

Ref. = Reference category; JHS = Junior High School; SHS = Senior High School; OR = Odds Ratio.

Nagelkerke R square = 0.200.

birth weight were 1.6 times more likely to be stunted relative to children with high birth weight (AOR = 1.59, 95% CI: 1.11–2.27, p = 0.011). Our results also showed that children from households with improved toilet facilities were 43% less likely to be stunted as compared to children from households that practice open defecation (AOR = 0.57, 95% CI: 0.37–0.88, p = 0.011).

## Factors associated with both wasting and stunting among children aged 6–23 months in Ghana

As shown in Tables 4 and 5, the factors associated with both wasting and stunting include being a male child, low wealth index, low birth weight and unimproved toilet facility. However, children with fever in the past 2 weeks, children of mothers with low education and children in households with unprotected source of drinking water are factors associated with only wasting.

## Discussion

This study investigated the influence of IYCF indicators, socio-demographic and economic-related factors, and water and sanitation on stunting and wasting, using nationally

**Table 5. Factors associated with stunting among children aged 6 to 23 months.**

| Variable | Stunting | | | |
|---|---|---|---|---|
| | Adjusted OR (AOR) | 95% Confidence Interval | | p-value |
| | | Lower | Upper | |
| **Child's age** | | | | |
| 6 to 11 | Ref. | | | |
| 12 to 17 | 1.26 | 0.78 | 2.04 | 0.348 |
| 18 to 23 | 3.01 | 1.97 | 4.59 | < **0.001** |
| **Child's sex** | | | | |
| Male | 1.80 | 1.28 | 2.52 | **0.001** |
| Female | Ref. | | | |
| **Wealth** | | | | |
| Low | 2.18 | 1.27 | 3.76 | **0.005** |
| Medium | 2.09 | 1.20 | 3.62 | **0.009** |
| High | Ref. | | | |
| **Child's birth weight** | | | | |
| Small | 1.59 | 1.11 | 2.27 | **0.011** |
| Average | 2.97 | 1.72 | 5.15 | < **0.001** |
| Large | Ref. | | | |
| **Type of toilet facility** | | | | |
| Open defecation | Ref. | | | |
| Unimproved | 0.99 | 0.60 | 1.64 | 0.984 |
| Improved | 0.57 | 0.37 | 0.88 | **0.011** |
| **Household head** | | | | |
| Male | 1.48 | 1.00 | 2.19 | **0.049** |
| Female | Ref. | | | |

Ref. = Reference category; OR = Odds Ratio.

Nagelkerke R square = 0.14.

representative datasets from the most recent MICS in Ghana. A total of 2,529 children aged 6 to 23 months met the inclusion criteria for the study, of which 12.1% were wasted and 15.9% stunted. These malnutrition rates are lower than that of other developing countries [44,45] but higher than the WHO minimum cut-off of 5% and 10%, respectively [39]. These high prevalence of malnutrition could be due to factors such as low economic status; inadequate water and sanitation conditions; low maternal literacy rates; fever resulting from opportunistic infections; and low birth weight; factors that are consistent with the findings of this study and other studies [46,47]. This study revealed that socio-demographic and economic factors such as wealth, child's sex, and birth weight as well as unimproved sanitary conditions were associated with both wasting and stunting.

Our results suggest that male children have a higher odd of wasting and stunting and low birth weight increases the odds of stunting and wasting among children. Contrary to another study [44], this present study's finding implies female children were less likely than their male counterparts to be stunted and wasted. This is in tandem with previous studies on similar national datasets in Ghana [14,48] and other parts of Africa [49,50]. The increased susceptibility of male children to undernutrition might be due to the fact that male children are expected to grow slightly more rapidly, and the increased expected rate is perhaps more easily influenced by nutritional inadequacies and other exposures [51]. Also, children born low birth weight may already be lagging and disadvantaged in terms of their growth, making them

prone to early developmental challenges including a compromised immune system and increased susceptibility to infections [52]. Increasing household wealth, on the other hand, reduces the odds of both wasting and stunting, emphasizing the role of poverty in malnutrition among children. Also, the results suggest that improving household sanitation reduces the odds of stunting and wasting among children. On the contrary, our results suggest that when the household head was male, the child had a higher odd of stunting, but this did not hold for wasting. Although we had no data on maternal empowerment index, our result may conform with the findings that female-headed households have better nutrition as women are more empowered to make decisions on the nutrition of the household [53], improving the nutrition of the household and children in the long-term. Further, our results suggest that maternal education may have beneficial effects on the short-term nutrition of the child [54], as maternal educational status was significantly associated with wasting but not stunting. The IYCF indicators including MMF, MDD and MAD, with prevalence rates of 39.4%, 25.9% and 13.7%, respectively, did not have significant associations with either stunting or wasting at the multivariable level. Their low prevalence rates in the present study are comparable to findings of recent studies in the same location [20,25], suggesting a need for improvement in IYCF practices, with a focus on the critical first 1000 days, in Ghana. Stunting reflects malnutrition of long-term origin, and these dietary factors were measured using 24-hour recall; hence it is plausible, as in some studies [19,55], that recent dietary factors may not be sensitive in predicting stunting. In this study, the main IYCF indicators were not associated with stunting, even at the bivariate level. A situation which suggests recent food intake in the past 24 hours may not necessarily reflect long-term consumption patterns [56]. The lack of association between stunting and MDD has been reported in other studies [55,57–59]. Although wasting is defined as a measure of malnutrition of recent onset, this study found a weak association between MDD and wasting at the bivariate level; the observed association did not remain when adjusting for other factors including child sex and birth weight, maternal education, water and sanitation, and household wealth index. It may be that MDD is less effective in determining undernutrition in contexts with low socio-economic status, and poor water and sanitary conditions with high rates of infections. In our study, the mean MDD score (3.37 ± 0.05) was well below the recommended threshold of $\geq 5$ out of the 8 food groups [40]. This could also imply that dietary inadequacies are more likely in our study population, but we could not ascertain that with the available data. Nevertheless, several studies have shown multiple dietary inadequacies among children aged 6–23 months in Ghana [60]. Dietary inadequacies have a negative implication for proper immune-system function in preventing and ensuring recovery from illness. The preceding together with poor water and sanitation partly explains the significant infection (fever) among the children in the last 2 weeks in our study. However, our finding contradicts a multi-country intervention study in Zimbabwe, Kenya and Bangladesh [61], where IYCF but not Water Sanitation and Hygiene (WASH) reduced the odds of stunting among children. That study concluded that combining IYCF interventions with WASH interventions would not reduce stunting more than implementing only IYCF. Conversely, a study in India [62] reported that interventions that combined both IYCF and sanitation would yield greater impact than each in isolation; this is consistent with findings published in the LANCET [29] where achieving 100% coverage of all the 10 IYCF interventions will only lead to 20% reduction in undernutrition and hence emphasizes the importance of combining both IYCF and WASH interventions. Highlighting the relevance of inter-sectoral collaboration if significant progress is to be made in reducing the vicious cycle of malnutrition.

Research works have shown that lack of adequate dietary intake alone cannot explain the global burden of malnutrition [63], and diet-inclined interventions alone may not be the panacea to resolving this global malnutrition challenge [29]. Inadequate sanitary conditions and

poor access to water have been recognized as increasingly complex determinants of malnutrition among children [64,65]. Among a significant proportion of Ghana's population, access to improved sanitation and protected water facilities is a challenge that may influence nutrition through several pathways. Three of such pathways have been postulated [66]; (1) diarrheal diseases—where the ingestion of pathogen-ridden human excreta from food/water inhibits gastrointestinal nutrient absorption that increases the susceptibility to opportunistic infections that are associated with diarrhoea, (2) environmental enteropathy—a sub-clinical gastrointestinal tract inflammatory condition caused by ingestion of faecal bacteria, which results in impaired intestinal immune function and increased intestinal permeability, and subsequent malabsorption of nutrients, and (3) nematode infections–as humans consume soil contaminated with worm eggs infested human excreta, these soil-transmitted helminth infestations, compete with their human host for scarce nutrients which results in growth retardation. As such, emphasis on improving access to water and good sanitary practices among children up to 24 months may help reduce the high malnutrition rates in Ghana.

A comprehensive approach that centres on access to healthcare services, enhanced pre- and post-natal outreach services in hard-to-reach communities, healthy food options, and improved water and sanitation infrastructure, are necessary to address these public health complications. By addressing these issues, stakeholders can promote healthy pregnancies, improve maternal and young child health outcomes, and ultimately, improve overall health and well-being.

## Strengths and limitations

The main limitation of this study is its inability to infer causality of its findings due to the inherent limitation of cross-sectional studies. Some of the variables included in this study were subjectively measured and could either be subject to recall bias or measurement error. For instance, the child's birth weight was based on mother's recall which may not be a precise method for determining a child's birth weight. Additionally, the IYCF indicators relied on mothers/caregivers to recall all foods consumed in the past 24 hours, which could also suffer from recall bias. However, the effects of such bias on the outcome variables could be minimal as we expect these biases in exposure variables to be equally distributed among exposed and non-exposed respondents, given the very large sample size. The fact that the explanatory variables accounted for 20% and 14% of the outcome variables meant that this study could not include all potential exposure variables in its analysis. A limitation attributable to the MICS not including an extensive number of nutrition-centred variables. In our analysis, robust survey analysis techniques were undertaken. Considering that the MICS data is nationally presentative, our findings can be generalized to all children aged 6–23 months in Ghana.

## Conclusion

Wasting and stunting rates were found to be higher than WHO-recommended thresholds. Contrary to our a priori expectations, IYCF indicators were not associated with either wasting or stunting. Rather unprotected water and unimproved toilet facilities, among other factors were better predictors of wasting. Our findings indicate that although dietary diversity is an important determinant of nutritional status, in certain contexts, other factors such as poor water and sanitation, low maternal education, wealth, fever, and child's birth weight may be better predictors of undernutrition. It is evident from our analysis that the diet-sanitation-nutrition nexus is complex and may warrant further investigations. These findings equally situate maternal education and equitable wealth creation, just as WASH, as integral tools that stakeholders could harness in achieving recommended childbirth weights and adequate nutritional status among infants and young children. Due to these findings, the combination of

nutrition-specific and nutrition-sensitive interventions to simultaneously address the multiple determinants of undernutrition is recommended in the Ghanaian nutrition context.

## Supporting information

**S1 Checklist. STROBE statement—checklist of items that should be included in reports of observational studies.**
(DOCX)

## Acknowledgments

The authors are grateful to the Ghana Statistical Service and UNICEF for giving access to the multi-indicator cluster survey data.

## Author Contributions

**Conceptualization:** Hammond Yaw Addae, Fusta Azupogo.

**Data curation:** Hammond Yaw Addae.

**Formal analysis:** Hammond Yaw Addae, Mohammed Sulemana, Taminu Yakubu, Ambrose Atosona, Rafatu Tahiru, Fusta Azupogo.

**Methodology:** Hammond Yaw Addae, Mohammed Sulemana, Taminu Yakubu, Ambrose Atosona, Rafatu Tahiru, Fusta Azupogo.

**Software:** Hammond Yaw Addae, Mohammed Sulemana, Taminu Yakubu, Ambrose Atosona, Rafatu Tahiru, Fusta Azupogo.

**Supervision:** Hammond Yaw Addae.

**Validation:** Ambrose Atosona, Rafatu Tahiru.

**Writing – original draft:** Hammond Yaw Addae, Mohammed Sulemana, Taminu Yakubu, Ambrose Atosona, Rafatu Tahiru, Fusta Azupogo.

**Writing – review & editing:** Hammond Yaw Addae, Mohammed Sulemana, Taminu Yakubu, Ambrose Atosona, Rafatu Tahiru, Fusta Azupogo.

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
