## [Decision Letter · Decision Letter 0]

13 Sep 2023

PONE-D-23-13762Low birth weight, household socio-economic status, water and sanitation are associated with stunting and wasting among children aged 6-23 months: results from a national survey in GhanaPLOS ONE

Dear Dr. Addae,

Thank you for submitting your manuscript to PLOS ONE. After careful consideration, we feel that it has merit but does not fully meet PLOS ONE’s publication criteria as it currently stands. Therefore, we invite you to submit a revised version of the manuscript that addresses the points raised during the review process.

Major Revisions

We look forward to receiving your revised manuscript.

Kind regards,

Verda Salman, PhD

Academic Editor

PLOS ONE

3. We note that Figure 2 in your submission contain copyrighted images. All PLOS content is published under the Creative Commons Attribution License (CC BY 4.0), which means that the manuscript, images, and Supporting Information files will be freely available online, and any third party is permitted to access, download, copy, distribute, and use these materials in any way, even commercially, with proper attribution. For more information, see our copyright guidelines: http://journals.plos.org/plosone/s/licenses-and-copyright.

Additional Editor Comments:

\\major Revisions Required

Reviewers' comments:

Reviewer's Responses to Questions

**Comments to the Author**

1. Is the manuscript technically sound, and do the data support the conclusions?

Reviewer #1: Yes

Reviewer #2: Yes

2. Has the statistical analysis been performed appropriately and rigorously? 

Reviewer #1: Yes

Reviewer #2: Yes

3. Have the authors made all data underlying the findings in their manuscript fully available?

Reviewer #1: Yes

Reviewer #2: Yes

4. Is the manuscript presented in an intelligible fashion and written in standard English?

Reviewer #1: Yes

Reviewer #2: Yes

5. Review Comments to the Author

Reviewer #1: Dear Writer,

I would like to provide some comments on the manuscript. Firstly, it is important to clarify that this study represents a secondary analysis. Additionally, I would like to inquire if any statistical assessments were conducted to identify potential confounders and interactions.

I kindly request that you address the suggested revisions and proceed with the resubmission accordingly.

Reviewer #2: Dear authors,

I had the pleasure of reading your paper. It was an interesting read and indeed this topic is very important. My comments are as follows.

1) How did you choose the variables? there are several research papers on the same topic that use a long list of independent variables as determinants of stunting and wasting. How did you choose them? Did you follow any variable selection methodology?

2) While it is interesting to have separate regressions for Stunting, wasting and one with double burden, I missed the relative interpretation of the results in the discussion. It would be nice to understand how result differ for two different types of undernutrition and what are the implications of those findings.

3) Did you consider underweight as the third indicator of undernutrition as well? it would be nice to know your reasoning for excluding it from the analysis.

4) You have explained prevalence of stunting and wasting in the discussion session. Since these are your main variables, I expected these percentages at the beginning of the results section. It would be nice if you could move the discussion on stunting and wasting prevalence to the beginning of the results section.

All the best.

6. PLOS authors have the option to publish the peer review history of their article (what does this mean?). If published, this will include your full peer review and any attached files.

Reviewer #1: **Yes: **Asma Abdul Malik Qureshi

Reviewer #2: No

---

## [Author Response · Author response to Decision Letter 0]

21 Sep 2023

Dear PLOS ONE Editor,

We are grateful to you and the reviewers for your time in improving our manuscript titled “Low birth weight, household socio-economic status, water and sanitation are associated with stunting and wasting among children aged 6-23 months: results from a national survey in Ghana” (PONE-D-23-13762). We have edited the manuscript to improve it as suggested by you and the reviewers. In the text following, we respond to each of the comments with references to the location in the main manuscript. 

Journal Requirements

2. Data Availability statement

3. We note that Figure 2 in your submission contain copyrighted images.

Authors’ response

1. Please, Authors have made the necessary edits to the manuscript in conformity with PLOS ONE style requirements. 

2. Data availability has been updated.

3. The Authors have not been successful at receiving feedback from the copyright holders. Although the figure was just for illustrative purposes only, we do acknowledge the ramifications moving forward. As such, Fig 2 and its caption on lines 356 and 357 has been expunged. Corresponding changes have therefore been made to line 351.

Review #1 comment 1

1. Firstly, it is important to clarify that this study represents a secondary analysis. 

Authors’ Response

Thank you for drawing our attention to this very important detail. In the abstract section, on lines 34 to 36 we have replaced 

“The 2017/18 Ghana Multi-Indicator Cluster Survey data was analysed” with

“This is a secondary data analysis of the most recent (2017/2018) Ghana Multi-Indicator Cluster Survey (MICS) datasets.” 

And added the new statement to the Materials and methods section on lines 117 and 118.

Review #1 comment 2

Additionally, I would like to inquire if any statistical assessments were conducted to identify potential confounders and interactions.

Authors’ Response

We first conducted a bivariate analysis to identify the significant factors associated with the outcomes and all factors which were significantly associated with the outcome variables were included in the multivariate analysis to produce adjusted odds ratios. Further, we included pair-wise interaction effects of the independent variables, but none of them was statistically significant. 

Reviewer #2: 

Dear Authors,

I had the pleasure of reading your paper. It was an interesting read and indeed this topic is very important. My comments are as follows.

Review #2 comment 1

1) How did you choose the variables? there are several research papers on the same topic that use a long list of independent variables as determinants of stunting and wasting. How did you choose them? Did you follow any variable selection methodology?

Authors’ response

Thank you for committing your time to review this manuscript.

Inasmuch as Authors agree that there are several independent variables associated with stunting and wasting in literature. We also acknowledge that such variables are context-specific. It is important to say that this research was a secondary analysis of multi-indicator cluster survey datasets that were collected by the Ghana Statistical Service and other relevant stakeholders. The datasets contained a predefined and exhaustive list of health and nutrition-related variables including those that were included in this study. 

As such, the inclusion of these variables was pre-defined before the conduct of this study by the primary data collectors. Additionally, after checking for the possible confounders from the literature, we conducted a bivariate analysis and included only variables that were statistically significant at p < 0.05 in the multivariate analyses. Our approach ensured we produced adjusted odds ratios for both outcomes modelled without including too many possible confounders which do not produce the model. 

Review #2 comment 2

2) While it is interesting to have separate regressions for Stunting, wasting and one with double burden, I missed the relative interpretation of the results in the discussion. It would be nice to understand how results differ for two different types of undernutrition and what are the implications of those findings.

Authors’ response 

For similarities in the determinants of both wasting and stunting and their implications, Authors have included the following in the Discussion section on lines 371 to 382.

“Our results suggest that male children have a higher odds of wasting and stunting and low birth weight increases the odds of stunting and wasting among children. Likewise, increasing household wealth reduces the odds of both wasting and stunting, emphasizing the role of poverty in malnutrition among children. Also, the results suggest that improving household sanitation reduces the odds of stunting and wasting among children. On the contrary, our results suggest that when the household head was male, the child had a higher odds of stunting, but this did not hold for wasting. Although we had no data on maternal empowerment index, our result may conform with the findings that female-headed households have better nutrition as women are more empowered to make decisions on the nutrition of the household [47], improving the nutrition of the household and children in the long-term. Further, our results suggest that low birth weight may have detrimental effects on the long-term nutrition of the child [48], as low birth weight was significantly associated with stunting but not wasting.” 

Review #2 comment 3

3) Did you consider underweight as the third indicator of undernutrition as well? it would be nice to know your reasoning for excluding it from the analysis.

Authors’ response

The Authors decided to focus on wasting and stunting because they are more relevant in the public health and clinical care settings in Ghana. Stunting is the most prevalent of the 3 three forms of malnutrition (with 1 in 3 children being stunted) and the acute nature of wasting makes it very relevant in the in-patient care environment. Also, Authors are of the view that underweight is a composite measure of both stunting and wasting, addressing the determinants of these two indicators meant addressing underweight in a nutshell. I am sure it is for this same reason that the sustainable development Goal 2.2 focused on only wasting and stunting. Concentrating on these 2 forms of undernutrition afforded Authors the opportunity to espouse a more focused narrative based on the study’s aims and scope.

Therefore for clarity, Authors added the below sentence to the Introduction, from lines 106 to 108.

“Limiting the outcome variables to only wasting and stunting is in tandem with the SDG [35] approach and mainly because underweight is a composite measure of both stunting and wasting.” 

Review #2 comment 4

4) You have explained prevalence of stunting and wasting in the discussion session. Since these are your main variables, I expected these percentages at the beginning of the results section. It would be nice if you could move the discussion on stunting and wasting prevalence to the beginning of the results section.

Authors’ response

I appreciate your suggestions and have therefore moved the statement below from the Results section from line 276 to line 270.

“The prevalence of wasting and stunting among the children was 12.1% and 15.9% respectively.” 

Authors are however unable to move the discussion of the prevalence of stunting and wasting from the discussion section to the Results section. This is due to the unique distinction between results and discussion sections as arranged by Authors.

Thank you, reviewer #2, for your patience.

All in-text suggestions for corrections have been effected were necessary, on lines 46, 79, 80, 106, 122, 123, 149 and 183. Additionally, specific changes have been made to the last row of Table 1 and 5th column of Table 2 as recommended.

Thank you for your time.

Sincerely yours, 

Addae Yaw Hammond

Corresponding Author

---

## [Decision Letter · Decision Letter 1]

3 Jan 2024

PONE-D-23-13762R1Low birth weight, household socio-economic status, water and sanitation are associated with stunting and wasting among children aged 6-23 months: results from a national survey in GhanaPLOS ONE

Dear Dr. Addae,

Thank you for submitting your manuscript to PLOS ONE. After careful consideration, we feel that it has merit but does not fully meet PLOS ONE’s publication criteria as it currently stands. Therefore, we invite you to submit a revised version of the manuscript that addresses the points raised during the review process.

**ACADEMIC EDITOR:  major revisions required.**One of the reviewers is not satisfied with the revisions. Please find below the comments of the reviewer and submit your response.==============================

We look forward to receiving your revised manuscript.

Kind regards,

Verda Salman, PhD

Academic Editor

PLOS ONE

Additional Editor Comments:

One of the reviewers is not satisfied with the revisions. Please find below the comments of the reviewer and submit your response.

Thank you for your detailed responses to my queries. Even though your responses are well thought out, I am still not convinced with some of them. I am going to explain my reasoning below.

1. Review #2 comment 1

I am fully aware that MICS has a predetermined set of variables for each round across countries. My concern was related to some commonly used variables that are available in MICS but missing from this analysis eg mother's media access, birth rank of the child, father's education, hand washing facility, agricultural land ownership etc. Why were these variables excluded?

Second paragraph of the author's response is much more problematic because I do not think that significance of bivariate correlations can be used for variable selection in multivariate analysis. Simplest methodology could be Backward Elimination starting from the most complete model. Current method is not correct and should be revised.

2.Review #2 comment 2

Thank you for adding the comparative analysis. Discussion on double burden is still missing.

3. Review #2 comment 3

While it is correct that underweight is a composite of stunting and wasting, the claim that it was excluded from SDGs for this reason needs a strong citation. Also, why does this reasoning disqualify underweight as an important indicator of undernutrition from your analysis?

4.Review #2 comment 4

I truly respect author's unique arrangement of arguments, but I am afraid uniqueness is not a good justification for any arrangement. Intuitive arrangement would explain the problem in detail (detailed analysis of prevalence of undernutrition) first and then move on to analyze the determinants of the problem. For me it is counterintuitive to discuss why a problem exists first and then discuss what the problem actually is.

Reviewers' comments:

Reviewer's Responses to Questions

**Comments to the Author**

1. If the authors have adequately addressed your comments raised in a previous round of review and you feel that this manuscript is now acceptable for publication, you may indicate that here to bypass the “Comments to the Author” section, enter your conflict of interest statement in the “Confidential to Editor” section, and submit your "Accept" recommendation.

Reviewer #2: (No Response)

2. Is the manuscript technically sound, and do the data support the conclusions?

Reviewer #2: Partly

3. Has the statistical analysis been performed appropriately and rigorously? 

Reviewer #2: No

4. Have the authors made all data underlying the findings in their manuscript fully available?

Reviewer #2: Yes

5. Is the manuscript presented in an intelligible fashion and written in standard English?

Reviewer #2: Yes

6. Review Comments to the Author

Reviewer #2: Dear authors,

Thank you for your detailed responses to my queries. Even though your responses are well thought out, I am still not convinced with some of them. I am going to explain my reasoning below.

1. Review #2 comment 1

I am fully aware that MICS has a predetermined set of variables for each round across countries. My concern was related to some commonly used variables that are available in MICS but missing from this analysis eg mother's media access, birth rank of the child, father's education, hand washing facility, agricultural land ownership etc. Why were these variables excluded?

Second paragraph of the author's response is much more problematic because I do not think that significance of bivariate correlations can be used for variable selection in multivariate analysis. Simplest methodology could be Backward Elimination starting from the most complete model. Current method is not correct and should be revised.

2.Review #2 comment 2

Thank you for adding the comparative analysis. Discussion on double burden is still missing.

3. Review #2 comment 3

While it is correct that underweight is a composite of stunting and wasting, the claim that it was excluded from SDGs for this reason needs a strong citation. Also, why does this reasoning disqualify underweight as an important indicator of undernutrition from your analysis?

4.Review #2 comment 4

I truly respect author's unique arrangement of arguments, but I am afraid uniqueness is not a good justification for any arrangement. Intuitive arrangement would explain the problem in detail (detailed analysis of prevalence of undernutrition) first and then move on to analyze the determinants of the problem. For me it is counterintuitive to discuss why a problem exists first and then discuss what the problem actually is.

Thank you

7. PLOS authors have the option to publish the peer review history of their article (what does this mean?). If published, this will include your full peer review and any attached files.

Reviewer #2: No

---

## [Author Response · Author response to Decision Letter 1]

9 Jan 2024

Dear PLOS ONE Editor,

We are grateful to you and the reviewers for your time in improving our manuscript titled “Low birth weight, household socio-economic status, water and sanitation are associated with stunting and wasting among children aged 6-23 months: results from a national survey in Ghana” (PONE-D-23-13762R1). The Authors have provided clarification on our statistical analyses and have provided references to justify the use of such analysis. We have also added more content to the discussion to fully address reviewer comments. In the following text, we provide further explanations to each of the comments with references to the location in the main manuscript. 

1. Review #2 comment 1

I am fully aware that MICS has a predetermined set of variables for each round across countries. My concern was related to some commonly used variables that are available in MICS but missing from this analysis eg mother's media access, birth rank of the child, father's education, hand washing facility, agricultural land ownership etc. Why were these variables excluded?

Second paragraph of the author's response is much more problematic because I do not think that significance of bivariate correlations can be used for variable selection in multivariate analysis. Simplest methodology could be Backward Elimination starting from the most complete model. Current method is not correct and should be revised.

Authors’ response

Authors would want to kindly remind the reviewer that we did not use bivariate correlations in any of our analyses. In the instance where we used bivariate analysis, the authors used bivariate regression analyses to first identify the variables that were significant at 0.05 p-value or less. The use of these bivariate associations to justify a variable’s inclusion in a multivariate analysis is a rigorous approach to statistical analyses and has been justified and used extensively in literature. 

The bivariate analysis is meant to serve as a screening tool, so only variables that produce the model will be included in the multivariate model. And this explains why some nutrition-related variables such as those listed by the reviewer may be present in the MICs dataset yet not included in our analysis. It becomes a challenge to include hundreds of variables in a multivariate model in the instance of a multi-indicator cluster survey. So, the need to first conduct this bivariate analysis before the multivariate.

Authors do however note the reviewer’s description of this approach as “problematic” and “not correct”. And would want to say that contrary to these statements, this approach is acceptable within this context. Due to its use in articles that have been published in this respected Journal and elsewhere, we are of the view that this approach is neither problematic nor incorrect.

For instance, in Abbas & Salman, 2023 (doi.org/10.1177/0192513X23120197) titled “Women’s Attitudes Toward Intimate Partner Violence in Sindh, Pakistan: An Analysis of Multiple Indicators Cluster Survey”, Abbas & Salman described the selection of their variables for inclusion in their multivariate analyses using this approach. 

In another article that was published on a similar dataset in this Journal, Sakpota & Hu (2023) (doi.org/10.1371/journal.pone.0287974) in their analysis reported that “First, bivariate analysis was conducted … to determine the p-value between diarrhea and independent variables, and variables having p-value less than 0.05 in bivariate analysis were considered candidate variables for the multilevel analysis”.

Authors have therefore provided a book by Hosmer & Lemeshow (Applied Logistic Regression, 2004) as citation [43] in the statistical analysis section to buttress the rigour of this approach of selecting variables for inclusion in a multivariate regression model. In Hosmer & Lemeshow’s book, which has been cited more than 77,000 times, they provide further rationale for conducting a bivariate analysis before a multivariate analysis. 

Please check line 230.

2. Review #2 comment 2

Thank you for adding the comparative analysis. Discussion on double burden is still missing.

Authors’ response

Thank you for drawing our attention to this. Authors have added the statement below to the discussion to cater for this omission.

“Contrary to another study [44], this present study’s finding implies female children were less likely than their male counterparts to be stunted and wasted. This is in tandem with previous studies on similar national datasets in Ghana [14,48] and other parts of Africa [49,50]. The increased susceptibility of male children to undernutrition might be due to the fact that male children are expected to grow slightly more rapidly, and the increased expected rate is perhaps more easily influenced by nutritional inadequacies and other exposures [51]. Also, children born low birth weight may already be lagging and disadvantaged in terms of their growth, making them prone to early developmental challenges including a compromised immune system and increased susceptibility to infections [52]. Increasing household wealth, on the other hand, reduces the odds of both wasting and stunting, emphasizing the role of poverty in malnutrition among children.”

Please check lines 364 to 376

3. Review #2 comment 3

While it is correct that underweight is a composite of stunting and wasting, the claim that it was excluded from SDGs for this reason needs a strong citation. Also, why does this reasoning disqualify underweight as an important indicator of undernutrition from your analysis?

Authors’ response

Thank you for requesting for more details on the reasons why Authors excluded underweight from the analysis in this present study. To provide further clarity Authors need to acknowledge the relevance of this study’s title and objectives. And that we followed and were guided by the title and objective of this study in conformity with established scientific literature.

The title of this manuscript reads. “Low birth weight, household socio-economic status, water and sanitation are associated with stunting and wasting among children aged 6-23 months: results from a national survey in Ghana”.

The objective of this study also reads; “…purpose of this study was to investigate the relationship between IYCF indicators, socio-economic, demographic, water and sanitation factors, and the outcome variables stunting and wasting among children aged 6 to 23 months.”

The objective and title were the overriding guide in conducting our analyses. And the title and objective as stated above are on wasting and stunting only. Moreover, the concentration of this study on these two indicators of malnutrition is an acceptable approach that conforms with established scientific literature.

For instance, studies published in this Journal have in the past concentrated on three indicators e.g. Boah et al., 2019 (doi.org/10.1371/journal.pone.0219665) or two indicators e.g. Mutanga et al., 2021 (doi.org/10.1371/journal.pone.0259765) or even one indicator e.g. Hossain et al., 2022 (doi.org/10.1371/journal.pone.0278097) based on their respective study titles and objectives. 

In a similar context as this present study, Mutunga et al., in 2020 (doi.org/10.3390/nu12020559) analysed the MICs of 6 countries. In that study, Mutunga et al. focussed on only stunting and wasting. These references may indicate that, although commonly practised, it is not mandatory for a study to analyse all 3 z-score anthropometric indices of child undernutrition. It is permissible to select one or two indices to be included in a study depending on what the study’s objectives are.

4. Review #2 comment 4

I truly respect author's unique arrangement of arguments, but I am afraid uniqueness is not a good justification for any arrangement. Intuitive arrangement would explain the problem in detail (detailed analysis of prevalence of undernutrition) first and then move on to analyze the determinants of the problem. For me it is counterintuitive to discuss why a problem exists first and then discuss what the problem actually is.

Authors’ response

Authors perfectly agree with the reviewer that “it is counterintuitive to discuss why a problem exists first and then discuss what the problem actually is.” The Authors would kindly want to point out that from the first submission of this manuscript to the revised submission, the authors have not done anything different. We first discussed the prevalence of wasting and stunting in the initial paragraphs of the discussion from lines 353 to 363. And in the ensuing paragraphs, we discussed the determinants of stunting and wasting form lines 364 to 433.

What Authors disagreed with was the reviewer’s initial (first) review suggestion for Authors to “…move the discussion on stunting and wasting prevalence to the beginning of the results section.” For which reason Authors stated that the discussion and results sections are uniquely different.

We are thankful to the Editor and the reviewers for their meticulousness and very significant inputs, hopefully, this time this manuscript will be suitable for publication in this reputable Journal.

Sincerely yours

Addae Yaw Hammond

Corresponding Authors

---

## [Editor Report · Decision Letter 2]

11 Jan 2024

Low birth weight, household socio-economic status, water and sanitation are associated with stunting and wasting among children aged 6-23 months: results from a national survey in Ghana

PONE-D-23-13762R2

Dear Dr. Addae,

We’re pleased to inform you that your manuscript has been judged scientifically suitable for publication and will be formally accepted for publication once it meets all outstanding technical requirements.

Kind regards,

Verda Salman, PhD

Academic Editor

PLOS ONE

Additional Editor Comments (optional):

Accepted for Publication
---

## [Editor Report · Acceptance letter]

19 Mar 2024

PONE-D-23-13762R2 

PLOS ONE

Dear Dr. Addae, 

I'm pleased to inform you that your manuscript has been deemed suitable for publication in PLOS ONE. Congratulations! Your manuscript is now being handed over to our production team.

Kind regards, 

on behalf of

Dr. Verda Salman 

Academic Editor

PLOS ONE